# Meta-analysis of Epstein-Barr virus genomes in Southern Chinese identifies genetic variants and high risk viral lineage associated with nasopharyngeal carcinoma

Ka Wo Wong[1], Kwai Fung Hui[2], Ki Pui Lam[3], Dora Lai-wan Kwong[4], Maria Li Lung[4], Wanling Yang[1], Alan K. S. Chiang[1] *

1 Department of Paediatrics and Adolescent Medicine, School of Clinical Medicine, Li Ka Shing Faculty of Medicine, The University of Hong Kong, Hong Kong SAR, China, 2 Department of Pathology, United Christian Hospital, Hong Kong SAR, China, 3 Division of Immunology, Boston Children's Hospital, Harvard Medical School, Boston, Massachusetts, United States of America, 4 Department of Clinical Oncology, School of Clinical Medicine, Li Ka Shing Faculty of Medicine, The University of Hong Kong, Hong Kong SAR, China

* chiangak@hku.hk

**Data Availability Statement:** The raw reads of the EBV sequencing data are deposited in NCBI database under BioProject ID PRJNA577485

## Abstract

Genetic variants in Epstein-Barr virus (EBV) have been strongly associated with nasopharyngeal carcinoma (NPC) in South China. However, different results regarding the most significant viral variants, with polymorphisms in EBER2 and BALF2 loci, have been reported in separate studies. In this study, we newly sequenced 100 EBV genomes derived from 61 NPC cases and 39 population controls. Comprehensive genomic analyses of EBV sequences from both NPC patients and healthy carriers in South China were conducted, totaling 279 cases and 227 controls. Meta-analysis of genome-wide association study revealed a 4-bp deletion downstream of EBER2 (coordinates, 7188–7191; EBER-del) as the most significant variant associated with NPC. Furthermore, multiple viral variants were found to be genetically linked to EBER-del forming a risk haplotype, suggesting that multiple viral variants might be associated with NPC pathogenesis. Population structure and phylogenetic analyses further characterized a high risk EBV lineage for NPC revealing a panel of 38 single nucleotide polymorphisms (SNPs), including those in the EBER2 and BALF2 loci. With linkage disequilibrium clumping and feature selection algorithm, the 38 SNPs could be narrowed down to 9 SNPs which can be used to accurately detect the high risk EBV lineage. In summary, our study provides novel insight into the role of EBV genetic variation in NPC pathogenesis by defining a risk haplotype of EBV for downstream functional studies and identifying a single high risk EBV lineage characterized by 9 SNPs for potential application in population screening of NPC.

## Author summary

Two genome-wide association studies (GWAS) of Epstein-Barr virus (EBV) in nasopharyngeal carcinoma (NPC) in South China have revealed different risk loci in EBER region

(https://www.ncbi.nlm.nih.gov/bioproject/?term=
PRJNA577485) and SRA ID SRP225584 (https://
www.ncbi.nlm.nih.gov/sra/?term=SRP225584).
The genotype information and the codes necessary
to reproduce the results of the 38 SNPs that
characterized the high risk EBV lineage for NPC in
South China are available on our GitHub repository
at https://github.com/AlanC-lab/Integrated_EBV_
genomes_NPC_South_China/.

**Funding:** This project was supported by funding
from the Health and Medical Research Fund of
Hong Kong Health Bureau, #02131706 and
#19180382 to AKSC. The funders had no role in
study design, data collection and analysis, decision
to publish, or preparation of the manuscript.
Website: https://rfs1.healthbureau.gov.hk/english/
funds/funds_hmrf/funds_hmrf_abt/funds_hmrf_
abt.html.

**Competing interests:** The authors have declared
that no competing interests exist.

and *BALF2* gene of EBV, respectively. More EBV genomes isolated from healthy donors
and NPC biopsies are included in this study. The meta-analysis of EBV genomes derived
from 279 NPC cases and 227 healthy controls identifies a risk haplotype of EBV compris-
ing of multiple genetic variants having strong linkage disequilibrium with each other. Fur-
thermore, a high risk lineage of EBV characterized by 9 single nucleotide polymorphisms
(SNPs) is found to be enriched in NPC of South China. The risk haplotype of EBV will be
useful for downstream studies of disease mechanisms and the high risk EBV lineage may
be included in screening protocol for early detection of NPC in the population.

## Introduction

Nasopharyngeal carcinoma (NPC) has an elevated incidence rate in Southeast Asia [1–3]. The
etiology of NPC is multifactorial with host genetics, environmental factors and viral factors
being identified as risk factors. NPC is more prevalent in male with a male-to-female incidence
rate ratio of 2.59 in 2019 [3]. The increased risk of NPC among family members of NPC
patients suggests a potential role for genetic predisposition in the development of NPC [4].
Genetic studies have proposed several susceptibility loci for NPC including genes in the HLA
locus and in the NF-κB pathway [5–12]. Exposure to carcinogens, such as preserved food and
cigarettes, is also associated with NPC [13,14]. Epstein-Barr virus (EBV) is one of the drivers of
the pathogenesis of NPC supported by the clonal expansion of EBV in infected tumor cells
[15].

The contribution of EBV genetic variation to NPC had been extensively studied. Early stud-
ies found that EBV could be broadly classified into Type I and Type II according to the
sequence variations in the *EBNA2*, *EBNA-3A*, *-3B* and *-3C* genes [16,17]. Globally, including
South China, Type I EBV is the most prevalent, except in Africa and Papua New Guinea where
Type I and Type II EBV genomes are equally abundant [18–23]. Previous study has shown
that NPC tumor cells predominantly harbor the Type I EBV strain [24]. EBV strains with an
extra BamHI restriction site inside the BamHI F fragment were more commonly found in
NPC tumor cells [25]. Viral oncogene *LMP1* with a mutation at the XhoI site in exon 1 and a
30-bp deletion in exon 3 were more common in endemic regions of NPC [26], but regional
case-control studies found that the frequency of this *LMP1* variant was not significantly higher
than that of the local controls [27]. The variant V-val of *EBNA1* also had an elevated frequency
in EBV genomes derived from NPC cases [28–30]. NPC-enriched *EBNA1* subtype that con-
tained the V-val variant could lead to a reduced ability to maintain the EBV episome [31].
Associations in *EBER* [32], *BZLF1* promoter [33] and *RPMS1* [34] were also reported to be
associated with NPC.

Recently, an increasing number of EBV genomes derived from clinical samples of NPC had
been sequenced. The advancement in sequencing technology has brought case-control studies
to a genome-wide scale. Currently, three EBV genome-wide association studies (GWAS) of
NPC in Southern Chinese had been conducted. Two studies included sporadic NPC samples,
one conducted in Hong Kong (designated as HK1) [21] and the other conducted in Guang-
dong and Guangxi (designated as GZ) [22]. The third GWAS comprised new familial NPC
samples collected in Guangdong (GZf) [23]. However, the three studies mapped the risk loci
to different EBV genomic regions. The HK1 study identified the susceptibility locus at *EBER2*
where the top signal is a 4-bp deletion downstream of *EBER2* (annotated as EBER-del) [21].
The GZf study identified the top variant of a 12-bp deletion upstream of the *BBLF2/BBLF3*
gene using familial NPC cases only and identified T6999G within *EBER2* to be the top variant

associated with NPC in the meta-analysis consisting of familial NPC cases and samples from HK1 and GZ studies [23]. The GZ study identified a lineage of NPC-dominant subgroup and proposed risk variants within the viral *BALF2* gene [22]. A recent EBV genomic analysis of NPC EBV genomes in Japan, a non-endemic region for NPC, demonstrated that the high risk loci in the *BALF2* gene are not prevalent in EBV genomes derived from Japanese NPC [35]. These findings support the association between high risk EBV variants and NPC in South China but detailed characterization of the high risk EBV variants is needed.

To further investigate the association between EBV genomic variations and NPC in South China, we set out to perform a meta-analysis of EBV genomes surveyed in Hong Kong, Guangdong and Guangxi. The GZf study samples are not included in this study due to non-availability of the genomic data. Apart from previously published EBV genomes, this study contributed new EBV genomes isolated from 61 biopsies of NPC and 39 saliva samples of healthy population controls of Hong Kong (designated as HK2). By combining the new data of HK2 with previous data of HK1 (combined dataset of HK1 and HK2 designated as HK) and GZ studies, a meta-analysis of GWAS was performed on the total number of 279 and 227 EBV genomes derived from NPC biopsies and control saliva, respectively. The results reproduced the association signals near the EBER region and identified a list of potential viral genetic variants in strong linkage disequilibrium to form a risk haplotype for NPC. Phylogenetic and population structure analyses further revealed that the high risk subtypes identified in previous studies might belong to the same high risk EBV lineage which can be characterized by specific single nucleotide polymorphisms (SNPs).

## Results

### EBV genomes isolated in Hong Kong and Guangdong/Guangxi provinces are closely related

In this study, we analyzed EBV genomes isolated from NPC cases and healthy donors from Hong Kong, Guangdong and Guangxi provinces of China. During the quality assessment phase, one EBV genome from the control sample in the first Hong Kong EBV GWAS study (designated as HK1) dataset was excluded in the downstream analyses due to a suboptimal genotyping rate (< 90%) (S1 Fig and S2 Table). Upon completing the quality control, the dataset for this meta-analysis comprised of EBV genomes obtained from 279 NPC biopsies and 227 control saliva samples. Among these, 62 NPC biopsies and 141 control saliva are derived from HK1, 61 NPC biopsies and 39 control saliva are newly sequenced and designated as HK2 and 156 NPC biopsies and 47 control saliva are derived from Guangdong and Guangxi provinces of China (designated as GZ) (Fig 1A and 1B). Ratio of gender and age between healthy donors and NPC cases is comparable (Fig 1C and 1D).

Guangdong and Guangxi provinces are situated approximately 100 and 600 kilometers, respectively, from Hong Kong. The Hong Kong Chinese population is closely related and genetically similar to the Chinese populations in Guangxi and Guangdong [22]. Principal component analysis (PCA) showed that different EBV types are distinguished by PC1 with Type I EBV genomes clustered on the left and Type II EBV genomes clustered on the right (Fig 2A). In comparison to the EBV genomes surveyed in Hong Kong, the GZ study captured fewer Type II EBV in the control set but more Type II EBV in the NPC set possibly due to a larger number of EBV genomes derived from NPC cases relative to the controls in their study. Overall, the EBV genomes presented in both Hong Kong and GZ studies are comparable and co-localize on the first two PCs (Fig 2A).

In order to verify the similarity of EBV genomes surveyed in the Hong Kong (HK) and GZ studies, a further PCA was conducted using a subset of samples that carried the Type I EBV

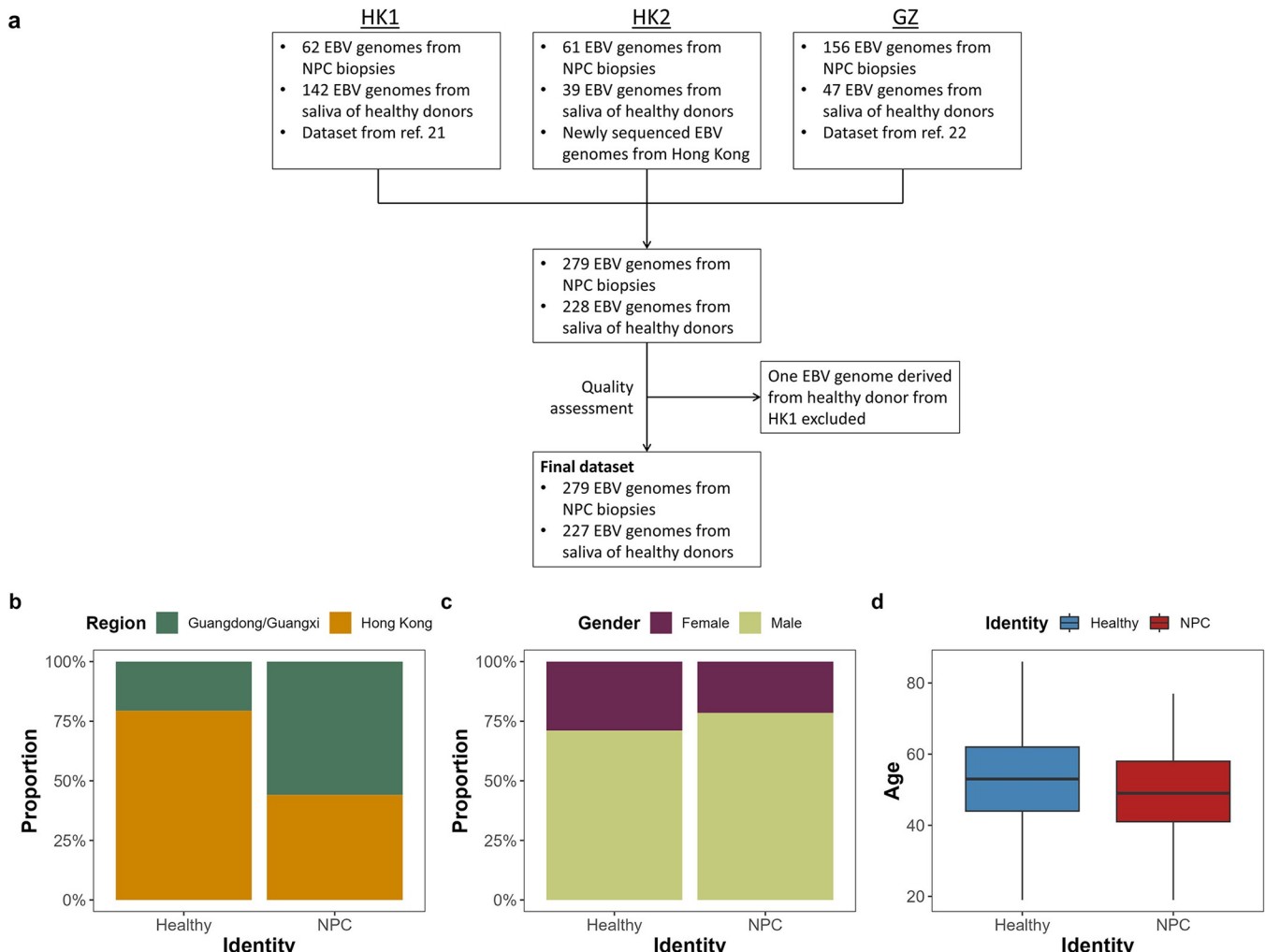

**Fig 1. Characteristics of samples in this study.** (a) Schematic diagram describing the composition of the dataset in this study. (b) Proportion of samples from Guangdong/Guangxi or Hong Kong in each cohort. (c) Proportion of female to male in each cohort. (d) Distribution of age in each cohort. GZ, dataset from reference [22]; HK1, dataset from reference [21]; HK2, dataset newly sequenced.

genome (n = 462). The smartpca algorithm identified three outliers which were subsequently excluded from the analysis. The plot of PC2 against PC3 did not reveal any clustering based on the dataset (Fig 2B). In addition, the calculation of pairwise nucleotide diversities among controls from HK and GZ as well as among NPC cases from HK and GZ aligned with the pairwise nucleotide diversity of their respective sampling locations. We also included a set of healthy control samples from Kenya to demonstrate the genetic similarity patterns among heterogenous samples. The pairwise nucleotide diversity between controls in South China and the Kenyan samples was notably high (indicating low genomic similarity) unlike the pairwise nucleotide diversity for HK and GZ. This finding indicates a high EBV genetic similarity across the two sampling sites in South China but not with the Kenyan samples (Fig 2C).

## GWAS replicates association signals near EBER2

To examine the potential relationship between EBV variant profiles and NPC, we conducted comparative analyses of sliding window and codon changes generated from healthy control

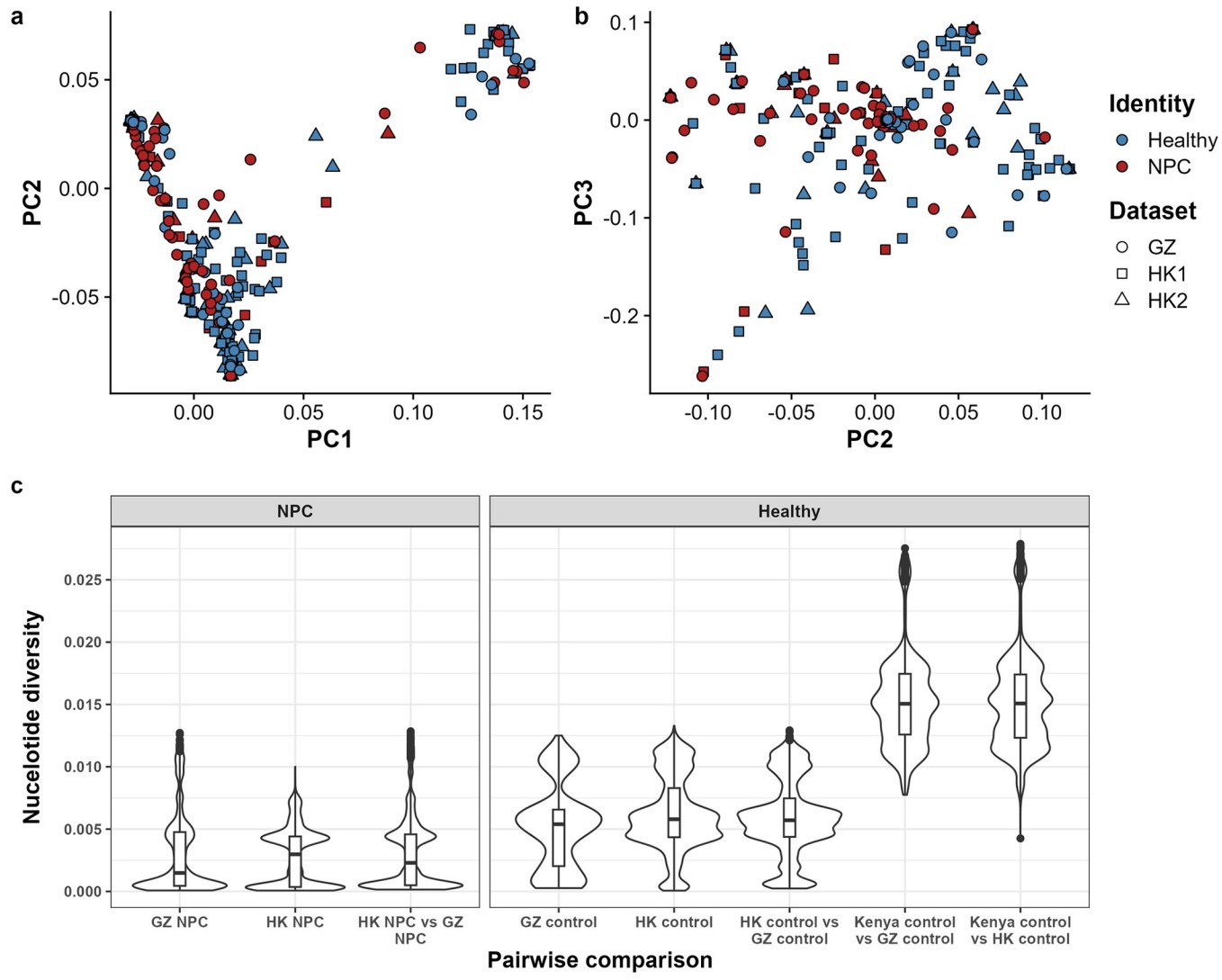

**Fig 2. Comparative analysis of EBV genomes in South China.** (a) First principal component (PC1) was plotted against the second principal component (PC2). (b) PC2 was plotted against PC3 calculated using samples of Type I EBV only (n = 459). Each shape represents one EBV genome, where different shapes indicate the dataset of where the sample originate. The colors indicate the identity of the sample (color and shape legends are shown on the right-hand side of the graph). GZ, dataset from reference [22]; HK1, dataset from reference [21]; HK2, dataset newly sequenced. (c) Analysis of pairwise nucleotide diversity among samples. The violin plot depicts the distribution of nucleotide diversity among each group of pairwise comparison. The panel on the left represents NPC cases and the panel on the right represents healthy controls. Additional samples from Kenya were included to show the nucleotide diversity pattern outside of South China. The lower the nucleotide diversity, the more similar the genomes.

and NPC cases, respectively. Our findings did not reveal any significant differences between NPC cases and controls in terms of variant profile and codon changes (S2 and S3 Figs). Subsequently, we conducted two separate GWAS on the EBV genomes surveyed in HK and GZ to investigate the relationship between NPC and common EBV variants with minor allele frequency >5%, variant missingness <10% and genotyping rate >90%. For the GWAS in HK, 123 NPC biopsies were compared against 180 controls and for the GWAS in GZ, 156 NPC cases were compared against 47 controls. A total of 1969 SNPs and 69 INDELs was tested in the HK GWAS while a total of 2021 SNPs and 78 INDELs was tested in the GZ GWAS. A logistic mixed regression model was employed for the GWAS. Age and sex of the individuals were included in the model as fixed effects and EBV genetic relatedness matrix was included

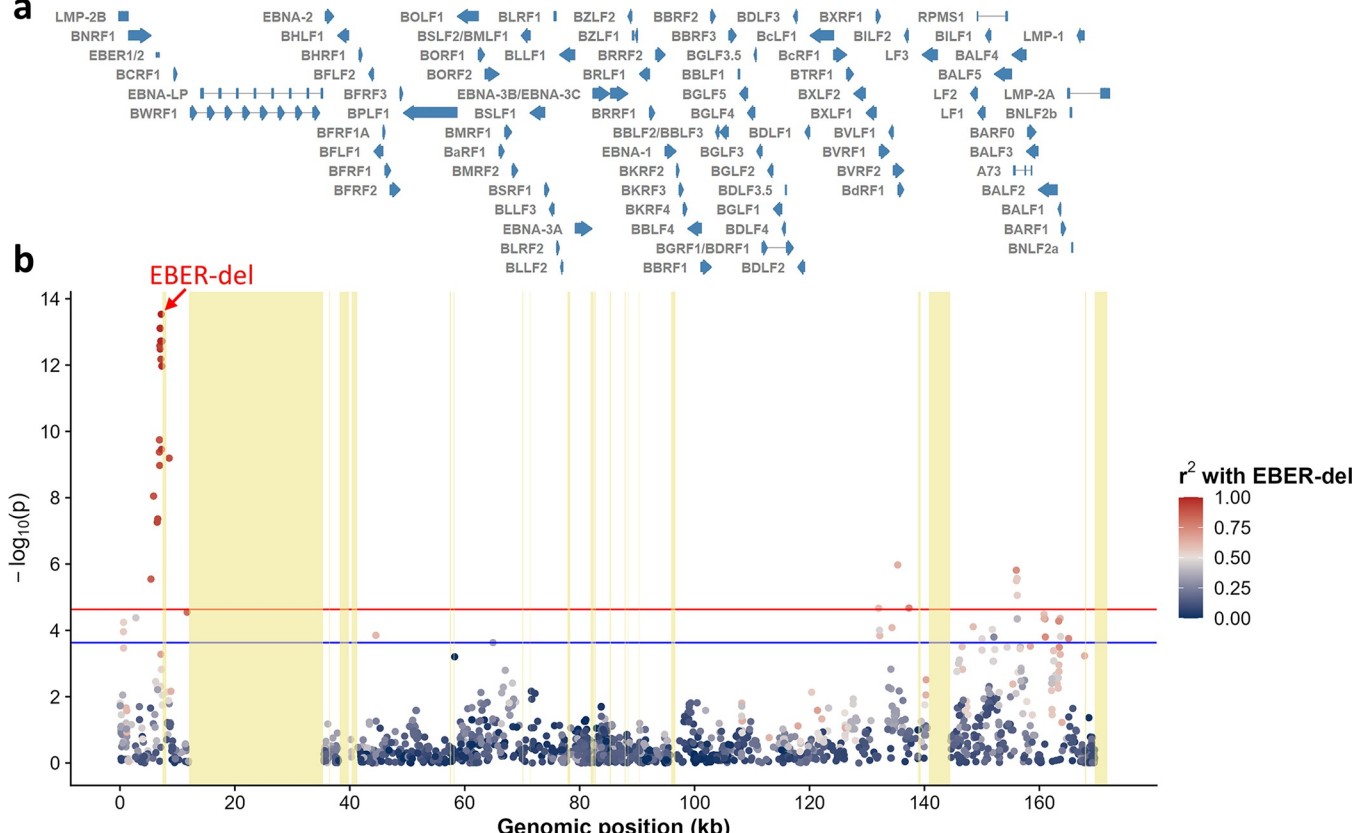

**Fig 3. Meta-analysis of EBV GWAS of NPC.** (a) EBV gene track annotated according to the Type I EBV reference (GenBank accession NC_007605.1). (b) Manhattan plot of the meta-analysis of GWAS. Variants are colored according to their linkage ($r^2$) with the top variant EBER-del downstream of EBER2, with red dots representing stronger linkage and blue dots representing weaker linkage. The top signal (EBER-del) in the meta-analysis is labelled with a red arrow. Repetitive regions in the EBV genome are shaded in yellow. Red line: Bonferroni-corrected genome-wide significance at $2.33 \times 10^{-05}$; blue line: permuted genome-wide significance at $2.35 \times 10^{-04}$.

as the random effect. We then pooled the results of the two GWAS for a meta-analysis where a total of 56 variants passed the permuted p-value threshold ($2.35 \times 10^{-04}$; Fig 3). Consistent with our previous study [21], the top association signal was located downstream of the viral non-coding RNA, EBER2, comprising of a 4-bp deletion (EBER-del) at coordinates, 7188 to 7191 ($p = 2.94 \times 10^{-14}$; case frequency = 0.918; control frequency = 0.427; S3 Table). Although the top risk variant discovered in the GZ study at coordinate 163364 in the *BALF2* gene ($p = 5.26 \times 10^{-05}$; case frequency = 0.885; control frequency = 0.427) also passed the permuted p-value threshold, the p-value ranked at 42[nd] among 56 variants that passed the threshold. We also observed that most of the variants that passed the permuted significance threshold were moderately to strongly linked to EBER-del including the variants near *BALF2* (Fig 3). The discrepancy between the findings of EBER-del and risk loci in *BALF2* was not attributed to the quality of called genotypes as demonstrated by minimal genotype missingness around the two loci (S4 Fig). We also investigated on the allele frequencies of the risk loci in *BALF2* and EBER-del between the two sampling locations (HK and GZ) and their differences were not statistically significant (S5 Fig).

Since a positive correlation between p-values of the meta-analysis (in negative log scale) and linkage disequilibrium (LD) with EBER-del was observed (Pearson's coefficient = 0.73; S6 Fig), we performed conditional analyses on EBER-del and variant 163364 separately followed

by meta-analyses on these conditional analyses of the two datasets (S3 Table). Our findings revealed that the significant signals in the 3' linear end of the EBV genome were dependent on EBER-del (S7A Fig) but not on variant 163364 (S7B Fig). This suggests that the variants in the EBER region may exert an effect on NPC independent of variant 163364.

## Identification of risk haplotype associated with NPC

Multiple variants have exceeded the genome-wide significance threshold in the meta-analysis and the existence of LD between EBER-del and these variants complicates the identification of pathogenic variant(s) for NPC. Therefore, we conducted a haplotype analysis formed by the 56 variants that reached the genome-wide p-value threshold (Fig 4A). After grouping low-frequency haplotypes together, we identified six haplotypes, namely haplotypes I-IV, mixed haplotype and risk haplotype. NPC cases were enriched in the risk haplotype (case frequency = 69.2%; control frequency = 28.6%; odds ratio = 5.59; p-value<$1.0\times10^{-09}$) which consisted entirely of the risk alleles identified in the meta-analysis of GWAS (S4 Table). Interestingly, haplotype III and the risk haplotype only differ in four sites (coordinates 148489, 150028, 151822 and 152087) but only 1.4% of NPC cases carried haplotype III in contrast to 69.2% for the risk haplotype. Similar to the testing of EBER-del and variant 163364, conditional analysis incorporating the above four sites as covariates did not demonstrate an

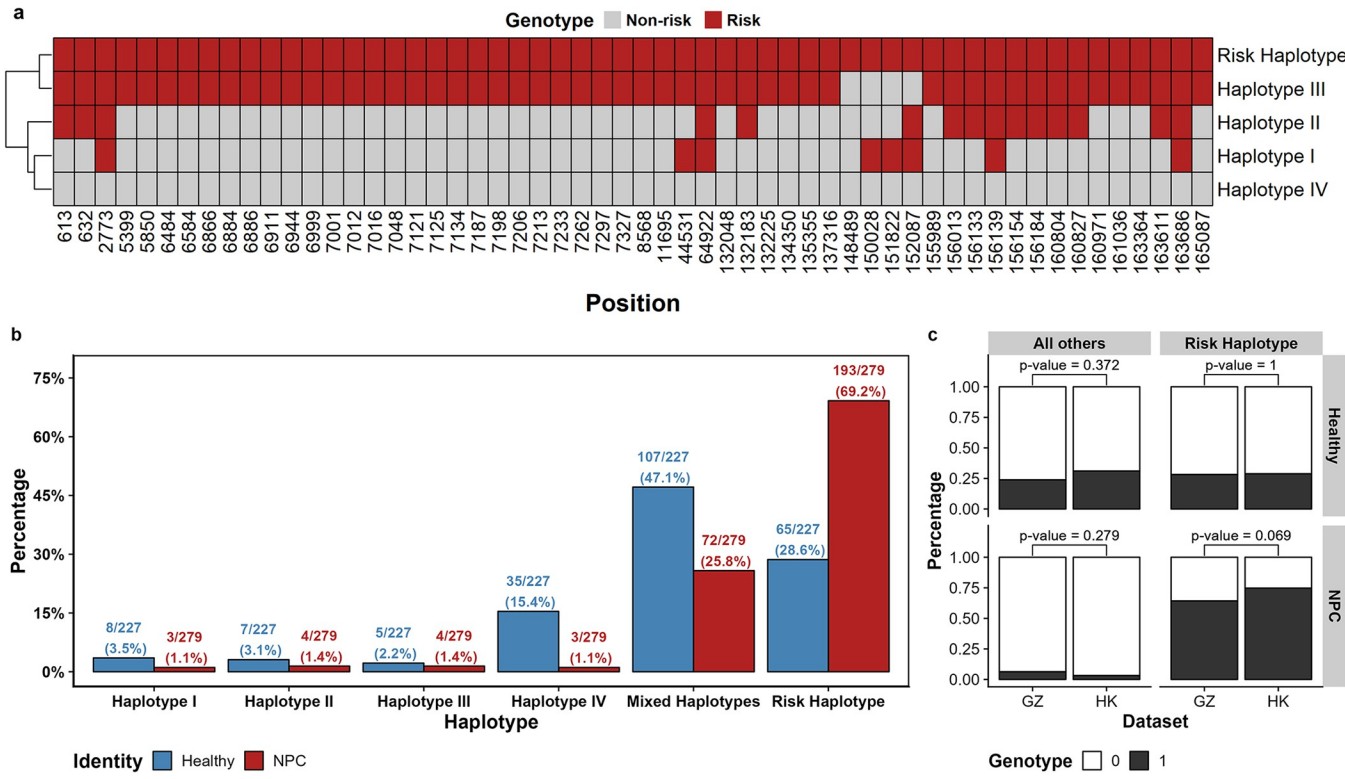

**Fig 4. Haplotype analysis of 56 variants that passed p-value cut-off in the meta-analysis.** (a) Heatmap showing the genotypes of the 56 variants in the five haplotypes with grey representing the non-risk allele and red representing the risk allele. The haplotypes are clustered according to their similarity, as shown by the dendrogram on the left-hand side. Mixed haplotypes are not shown. (b) Bar chart showing the percentages of NPC and healthy samples that carried the haplotypes with number of samples carrying the haplotype shown on top of each bar. Any haplotypes with less than seven individuals carrying it are grouped under mixed haplotypes. (c) Distribution of samples carrying the risk haplotype and other haplotypes, stratified by sample identity (Healthy/NPC) and sampling location (GZ/HK). The first row of the panel represents controls and the second row represents NPC cases. The first column represents haplotypes other than risk and the second column represents risk haplotype. P-values from the Fisher's exact test of haplotype counts are shown on top of each bar.

independent effect associated with NPC (S7C Fig). Most of the healthy individuals carried low-frequency haplotypes that were grouped together under "mixed haplotypes" (47.1%) and haplotype IV (15.4%) (Fig 4B). The haplotype frequencies in NPC cases and controls with respect to sampling locations (HK and GZ) are similar indicating no bias was observed between the two locations (Fig 4C). The haplotype analysis suggested that NPC might be associated with multiple EBV variants constituting a risk haplotype instead of a single variant. The risk haplotype should provide the basis for functional studies of pathogenic EBV variants.

## High risk EBV lineage is enriched in NPC

In addition to the high risk variants identified through genome-wide association tests, the HK and GZ studies have independently reported that certain EBV subpopulations are preferentially enriched in NPC. In this study, we created a phylogenetic tree and conducted a population structure analysis to identify the high risk EBV lineage of NPC. The phylogenetic tree revealed a lineage (Fig 5A) that was enriched with highly similar NPC EBV strains, confirming the equivalence of the high risk subpopulations identified in the HK1 study (subgroup 1B) [21] and GZ study (high risk subtype) [22]. Despite the NPC-associated variants being mapped to different regions in different studies, both studies concurred on the lineage level. To further characterize the subpopulations of the EBV genomes, we partitioned the dataset into clusters of similar genomes using genome-wide SNPs by the R package rhierBAPS [36]. A total of nine clusters were reported, and we observed that most of the EBV genomes derived from NPC are found in clusters 2, 3 and 7 (Fig 5B and S5 Table).

## The high risk EBV lineage is characterized by specific SNPs

To identify the SNPs that characterize the high risk EBV lineage, we identified the most common SNPs at each EBV coordinate of each cluster. Only the SNPs that were common to clusters 2, 3 and 7 (comprising the high risk lineage) were selected. In total, the high risk lineage can be characterized by a block of 38 SNPs consisting of loci from *BNRF1*, *EBERs*, *OriP*, *EBNA1*, *BGRF1/BDRF1*, *BcRF1*, *BVRF2*, *BALF5*, *BALF2*, *BARF1* and *LMP1*. Since multiple variants shared identical genotypic information across the 506 samples ($r^2 = 1$), these variants

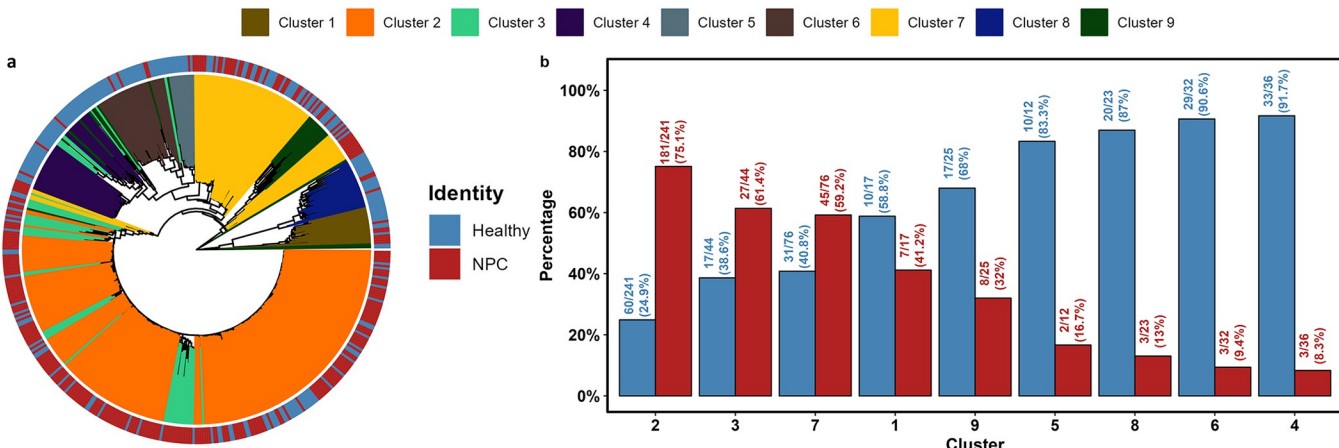

**Fig 5. Phylogenetic and population structure analysis.** (a) The phylogenetic tree was constructed from the multiple sequence alignment of their EBV genomes for all samples. The tree is rooted at the midpoint and each tip of the tree represents one sample. The colors of the lines extending from the tree branches indicate the clusters the samples belong to, which is shown on top of the figure. The outer ring represents the identity of the sample. (b) The percentages of NPC and healthy samples in each cluster. The numbers of NPC and healthy samples over the total numbers of individuals in each cluster are labelled on top of each bar, ordered in descending order of NPC to control ratios.

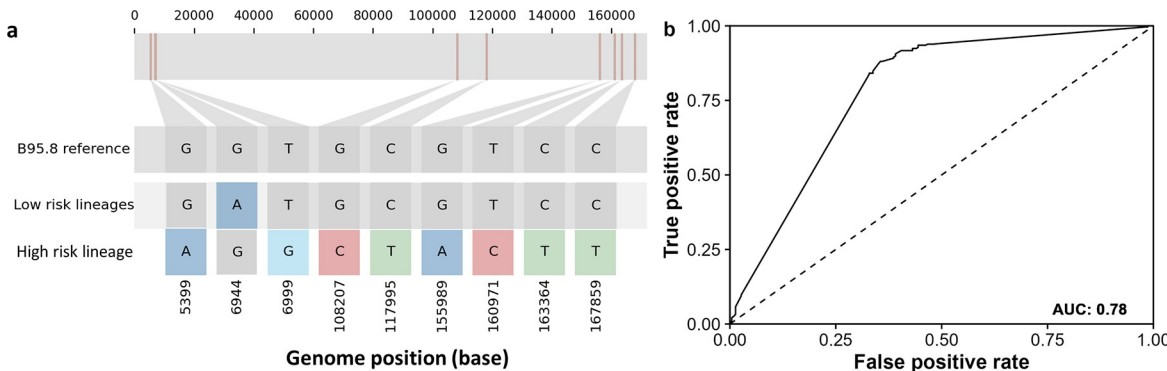

**Fig 6. SNP profile characterizing high risk EBV lineage.** (a) SNP profile including multi-allelic variants that characterizes the EBV high risk lineage. The genome tract on top represents the relative position of the SNPs in the EBV genome. The first panel of the annotation with grey squares denotes the reference sequence of B95.8, and the subsequent annotations with colored squares denote SNPs at various loci of low risk lineages and the high risk lineage, labelled at the bottom of the graph. (b) Receiver operating characteristic (ROC) curve showing the performance of the SNPs profile in predicting NPC. Area under the curve (AUC) is shown on the bottom-right corner.

were removed from the dataset reducing the number of SNPs to 28 (S6 Table). We further narrowed down the panel of 28 SNPs by combining the process of LD clumping and XGBoost to 9 SNPs (Fig 6A). All sites in the high risk lineage carried the alternative allele except position 6944 (12 bp upstream of *EBER2*) which carried the G genotype from the Type I reference genome. A total of 273 individuals out of 506 carried this panel of 9 SNPs (S8 Fig and S7 Table). The majority of these 273 represent NPC cases suggesting strong association between the panel of 9 SNPs and NPC (205/273; odds ratio = 6.48; Fisher's Exact p = $2.2\times10^{-16}$). Most of the individuals that carried the panel of 9 SNPs were classified in clusters 2, 3 and 7 (the high risk lineage) except for one NPC case in cluster 1 (S7 Table). Incorporating age and sex as covariates in a logistic model, the panel of 9 SNPs can effectively predict NPC cases with an area under the curve (AUC) value of 0.78 based on the receiver characteristic curve (ROC) (Fig 6B). The performance metrics in predicting NPC using 28 SNPs and 9 SNPs are comparable with the panel of 9 SNPs having slightly higher balanced accuracy. Hence, the panel of 9 SNPs can achieve similar screening efficiency for NPC with fewer markers (S9 Fig). Overall, NPC is associated with a high risk EBV lineage that is characterized by a set of variants. Our data support that the EBV lineage may be screened efficiently using a minimal set of 9 markers.

## Discussion

This study reports an analysis of three datasets of EBV genomes (HK1, HK2 and GZ) in order to resolve the association between EBV genetic variants and NPC in South China. Although previous studies with smaller sample sizes were able to yield significant GWAS results, they reported the main associated variants at *EBER2* region and *BALF2*, respectively [21–23]. The meta-analysis conducted by Zhang et al. identified a SNP of T6999G within *EBER2* to be the top variant associated with NPC [23]. The meta-analysis of GWAS conducted here replicated the association between variants near *EBER2* and NPC with the strongest signal coming from a 4-bp deletion downstream of *EBER2* (EBER-del). Moreover, several variants including those in the *BALF2* gene are strongly linked to the variants near EBERs and are strongly associated with NPC when the effect of EBER-del is not considered. Importantly, we confirmed that the high risk subpopulations described in different studies are, in fact, referring to the same high risk lineage characterized by variants near *EBER2*, *BALF2* and other linked variants.

EBER2, a non-coding RNA, is known to bind to multiple cellular factors to exhibit a myriad of functions in EBV-infected cells including the suppression of latent gene expression and modulation of cytokine secretion [37–41]. Recent studies investigated the functional role of the EBV strain M81 which is derived from a Southern Chinese NPC. The M81 EBV strain could induce the expression of the chemokine, CXCL8 and, in conjunction with the recognition of viral single-stranded RNA by toll-like receptor 7, might lead to chronic inflammation and drive virus lytic production [42]. The T6999G variant of *EBER2* discovered in the meta-analysis of familial NPC cases was reported to be among the most important SNPs that influence EBER2 expression [42] and establishment of latency [43]. In addition to the *EBER2* variants, several EBV genetic variants may also be associated with NPC independently and different GWAS may report varying sets of risk variants such as variants in *BALF2*. However, upon applying additional GWAS conditioning on the effect of EBER-del, the signals in other viral genetic regions were abolished suggesting their effects are largely dependent on the EBER region. Although we observed the most significant effect at EBER-del, the possibility that NPC is associated with multiple variants instead of a single one cannot be excluded. Therefore, a haplotype analysis based on all variants that passed the genome-wide p-value cut-off was carried out to investigate the association between these variants and NPC. We found that NPC cases were highly associated with a risk haplotype suggesting that multiple variants might contribute to NPC collectively. The variants comprising the high risk haplotype could reveal potential candidates for further investigation into their impact on NPC pathogenesis through functional studies.

Apart from identifying the casual viral variants for NPC, considerable efforts have been dedicated to the screening of NPC using plasma EBV DNA load where robust sensitivity and specificity are attained [44]. However, relying solely on plasma EBV DNA load for NPC screening may lead to false positive signals. To address this issue, the same group of researchers defined a set of 661 independent EBV single nucleotide variants (SNVs) to augment the NPC screening program [45]. Of note, the plasma EBV DNA levels of most individuals are suboptimal for sequencing and will pose significant challenge to accurately acquire risk genotypes across the EBV genome. In this study, we have identified a block of 9 closely linked SNPs primarily concentrated in the two linear ends of the EBV genome to characterize the high risk EBV lineage in NPC. By reducing the number of screening targets to 9 sites, a higher success rate in screening and genotyping all sites might be attainable. Incorporating this block of 9 SNPs of EBV in the screening protocol of NPC using plasma EBV DNA can potentially be verified in future studies.

In summary, our study provides novel insight into the role of EBV genetic variation in NPC pathogenesis by identifying a risk haplotype of EBV for NPC and a high risk EBV lineage characterized by 9 SNPs. These findings have significant implications for both functional studies of NPC pathogenesis and the development of population screening program for NPC in endemic regions.

## Materials and methods

### Ethics statement

The collection of the new samples received approval from the Institutional Review Board of The University of Hong Kong/Hospital Authority Hong Kong West Cluster (UW 08–156, UW 10–018 and UW 14–546). Written informed consents were obtained from the participants.

### Subject recruitment

The recruitment criteria of healthy population donors were described previously [21]. For the NPC samples in the HK2 dataset, we obtained 61 NPC biopsies with histologically confirmed

primary undifferentiated NPC from the NPC Tissue bank of the Centre for Nasopharyngeal Carcinoma Research (CNPCR) of The University of Hong Kong (HKU) and an established NPC tumor bank of our laboratory. The characteristics of the population carriers and NPC cases are shown in S1 Table.

## EBV genome dataset

The data analyzed in this study consisted of three datasets: HK1 [21], HK2 and GZ [22] of South China. The HK1 dataset consisted of 142 EBV genomes derived from the saliva of healthy donors and 62 EBV genomes derived from biopsies of NPC cases. The GZ dataset consisted of 47 EBV genomes derived from the saliva of healthy donors and 156 EBV genomes derived from biopsies of NPC cases. In the HK2 dataset, the 61 biopsies DNA collected from NPC cases were obtained from the NPC Tissue bank of the CNPCR at HKU. The 39 EBV genomes derived from control saliva were obtained from the samples that were not sequenced in the recruitment described in the HK1 study [21]. The participants in HK1 and HK2 are Hong Kong Chinese living in Hong Kong whereas those in GZ are Chinese living in Guangdong or Guangxi provinces of China. The raw reads from the HK1 and GZ dataset were downloaded from the NCBI SRA database with ID number SRP152584 and NCBI database with BioProject ID PRJNA522388, respectively. The sample information is listed in S1 Table.

## Sample preparation

DNA were extracted using AllPrep DNA/RNA microKit and Qiagen Blood and Tissue Kit according to the manufacturer's protocol (Qiagen, Hilden, Germany). The DNA concentrations were determined using NanoDrop spectrophotometer (Thermo Scientific) and Qubit dsDNA High Sensitivity (HS) Assay Kit (Life Technologies). The viral copies were quantified by qPCR using the FAM channel of the Applied Biosystems 7900HT Fast Real-Time PCR System (AppliedBiosystems, Life Technologies, CA). The viral load was quantified by targeting the EBV BamHI-W repeat region, with the forward primer (Integrated DNA Technologies) sequence 5'-GGTCGCCCAGTCCTACCA-3'; reverse primer (Integrated DNA Technologies) sequence 5'-GCTTACCACCTCCTCTTCTTGCT-3'; and TaqMan probe (Thermo Scientific) sequence 5'-CCAAGAACCCAGACGAGTCCGTAGAAGG-3'. The average of three technical replicates were reported. To ensure optimal sequencing depth, samples with viral load $>3 \times 10^4$ copies/ug DNA were selected for Illumina sequencing.

## Library preparation, target capture and sequencing

For each sample, 120 ng DNA was aliquoted in 60 ul 0.1x Tris-EDTA buffer (1 mM Tris-HCl, pH 8.0, 0.1 mM EDTA) (Life Technologies) in microtube AFA Fiber Snap-Cap 6 × 16 mm Case (Covaris). The tube of DNA was subjected to fragmentation by sonication to a target size of 350-400bp using the M220 Focused-ultrasonicator (Covaris) with a duty factor of 10%, 200 cycles per burst, peak incident power at 50W, and treated for two minutes. The size of the DNA fragments was checked on Agilent 2100 Bioanalyzer (Agilent) using the DNA high sensitivity chip, performed by the Centre for PanorOmic Sciences (CPOS) of HKU. The fragmented DNA were then prepared for Illumina sequencing using NEBNext Ultra II DNA Library Prep Kit for Illumina (New England Biolabs) according to manufacturer's instruction. The DNA library was enriched for EBV genome with a target capture method described previously [21]. The DNA libraries were sequenced (100 bp paired-end) by CPOS of HKU using the HiSeq 2500 Sequencer (Illumina).

## Read alignment, variant calling and filtering

The qualities of the raw sequencing reads were first checked using FastQC, which indicates good overall sequencing quality (overall mean rate $\geq$Q30 = 93.0% before filtering). The reads were then processed using fastp with the default settings to remove low-quality and low-complexity regions, and to remove Illumina adapter sequences. Processed reads were mapped to the Type I EBV reference genome (GenBank accession number: NC_007605.1) using BWA-MEM (version 0.7.17) [46]. The reads mapped to EBV were then sorted by the leftmost coordinates in SAMtools (version 1.14) [47]. Duplicated reads in the alignment were marked with the Picard toolbox (version 2.0.1). The base quality scores of the duplicates-marked-aligned reads were recalibrated using the base quality score recalibration (BQSR) tools in Genome Analysis Toolkit (GATK, version 4.3.0.0) based on our in-house EBV known site variant call format (VCF) file. Then, SNPs and short INDELs were called by the HaplotypeCaller under GATK to generate global VCF files (gVCF). The gVCF files were subsequently filtered, and the gVCF files across multiple studies were combined following GATK's best practice. Variants that lied in and within five base pairs of the EBV genome repetitive regions were masked from this study and were not analyzed. Samples with a genotyping rate less than 90% and with a sequencing depth excluding repetitive regions less than 15-fold were removed from the dataset. All samples reached the sequencing depth threshold of 15-fold, and one sample was excluded from the analyses due to a suboptimal genotyping rate.

Our final dataset consisted of 227 healthy individuals and 279 NPC cases. The EBV variants were annotated using snpEff (version 5.0e) [48] according to NC_007605.1, and the linkage disequilibrium (LD) statistics of all variants in the EBV genomes were calculated using PLINK (version 1.90b6.22) [49]. To determine the EBV type of our samples, we aligned the Type II EBV reference genome AG876 (GenBank accession number: NC_009334.1) to the Type I EBV reference genome generating an EBV Type II BAM file. The BAM file for each sample was then combined with the Type II EBV reference BAM file using the mpileup function in BCFtools (version 1.13) to generate genotype likelihood at each genomic position. Variants were subsequently called by BCFtools including only multiallelic SNPs. Nucleotide similarity per 1000 base-pair window with a step size of one base-pair was calculated and the result was plotted in a graph. EBV typing was performed manually by checking the peaks in each similarity graph.

## Principal component analysis (PCA)

To perform PCA, SNPs with missingness less than 0.1 and minor allele frequency greater than 0.05 were selected. The PCA was implemented using the smartpca algorithm in the EIGEN-STRAT software (version 6.1.4) with the default parameters, where outliers were removed in five iterations if they are more than six standard deviations away from the top ten PCs [50].

## Genome-wide association tests (GWAS), meta-analysis of GWAS and haplotype analysis

Variants with missingness less than 0.1 and minor allele frequency greater than 0.05 were retained. The logistic mixed model was implemented in the GMMAT package (version 1.4.0) in R [51], running it separately for HK (180 controls and 123 NPC cases) and GZ (47 controls and 156 NPC cases) samples. Age and sex were included as fixed effects, while an EBV genetic relatedness matrix, generated by GEMMA (version 0.98.5) [52], was incorporated as the random effect. We used the Wald test to derive the effect sizes and p-values. In the conditional analyses, we included the genotypes of the target variant for each individual, along with age and sex, as fixed effects.

The meta-analysis was based on the two GWAS summaries of HK and GZ samples, implemented in METAL (version 2011-03-25) [53]. The p-values reported from the two GWAS were weighted by the number of individuals contributing to each marker in the meta-analysis. To control for type I error due to multiple testing, we applied two p-value cut-offs. First, we employed the Bonferroni correction with an alpha level of 0.05, resulting in a genome-wide p-value threshold of $2.33×10^{-05}$. Secondly, we determined an additional p-value cut-off using label-swapping permutation tests. In this process, we randomly shuffled the phenotype labels (case or control) within each dataset, and each individual was assigned to one phenotype label. We repeated this process for one thousand times, conducting a GWAS and meta-analysis of GWAS in each iteration. The lowest p-value from the meta-analysis in each iteration was recorded. After all iterations, the p-values were sorted from the smallest to the largest, and the p-value at the 5$^{th}$ percentile was selected as the permuted p-value cut-off, which is $2.35×10^{-04}$.

All variants that passed the permuted threshold in the meta-analysis were selected for haplotype analysis, resulting in 54 SNPs and two INDELs. The haplotype analysis was conducted in the haplo.stats package (version 1.9.3) in R. A logistic regression model was constructed using the haplo.glm function. This model was built on the 56 variants against a binary phenotype (comprising 227 controls and 279 NPC cases) of individuals. Age and sex were also included in the model as covariates.

## Phylogenetic analysis and population structure analysis

To build a phylogenetic tree, individual FASTA files representing the EBV genome were generated from their corresponding VCF files. This was done in accordance with the EBV reference genome NC_007605.1, with nucleotides in the repetitive regions designated as "N". Then, a multiple sequence alignment of all the FASTA sequences was created using MAFFT (version 7.505) [54]. Subsequently, a maximum likelihood tree was constructed using IQ-TREE (version 2.0.7). The ModelFinder Plus utility was employed to select the best-fit model based on the Bayesian information criterion [55]. The chosen model for the phylogenetic tree was GTR +F+R3, which is a general time reversible model with unequal rates and unequal base frequencies. The resulting tree was rooted at the midpoint and visualized using the ggtree package (version 3.8.0) in R [56]. On the other hand, a core SNP alignment was clustered using the rhierBAPS package (version 1.1.4) in R with the default setting [36] for population structure analysis.

## Identification of single nucleotide profile that characterized the high risk EBV lineage

The SNPs that characterized the high risk lineage were identified with an in-house R script. In brief, for each genomic position within each population cluster, the most common non-missing SNP was curated. If multiple alleles were equally abundant at a certain site, these alleles were concatenated and considered together. Subsequently, clusters 2, 3 and 7 were labelled as "high risk", while clusters 1, 4, 5, 6, 8 and 9 were labelled as "low risk". The most common allele per position was then aggregated according to these labels. Only positions with alleles unique to the "high risk" label were selected resulting in a total of 38 SNPs. The genotype information and the codes necessary to reproduce the results are available on our GitHub repository at https://github.com/AlanC-lab/Integrated_EBV_genomes_NPC_South_China/.

To further refine our results, we first removed SNPs that shared identical genotype information across the 506 samples, resulting in 28 SNPs. We then employed LD clumping in PLINK and XGBoost feature selection algorithm in the xgboost package (version 1.7.7.1) in R to further analyze these 28 SNPs. In the LD clumping step, we ranked the 28 SNPs based on

their p-values reported in the meta-analysis of GWAS, then selected one SNP with the lowest p-value per group of SNPs that was in high LD with each other. An $r^2$ threshold of 0.75 for the LD clumping was applied yielding six SNPs. Simultaneously, we used XGBoost, a machine learning algorithm, to rank the importance of the SNPs based on their predictive power for high risk EBV lineage. We trained the XGBoost model using a binary classification on disease status (healthy or NPC) and the 28 SNPs were included as features in the model. After training the model, we selected the top five features based on their Gain scores from XGBoost. The resulting SNPs from LD clumping and XGBoost were combined, forming a total of 9 SNPs (two SNPs overlapped between XGBoost and LD clumping). The combined features were fitted into a logistic regression model to predict for NPC followed by evaluation with ten-fold cross-validation with five-repeats using the caret package (version 6.0–94) [57] in R.

## Statistical analysis

The Fisher's exact test was employed to test the differences in allelic and haplotypic counts between HK and GZ datasets. All statistical analyses were conducted in R (version 4.3.0) unless otherwise specified.

## Supporting information

**S1 Fig. Genotyping rate and sequencing depth of 507 EBV genomes.** Each dot represents the genotyping rate of a sample. Horizontal red line indicates the genotyping rate threshold at 0.9.
(TIF)

**S2 Fig. Sliding window analysis with a window size of 1000 nucleotides and a step size of one base-pair.** Blue and red lines indicate the variant profiles for healthy individuals and NPC cases respectively. Repetitive regions in the EBV genome are shaded in yellow.
(TIF)

**S3 Fig. Number of synonymous and non-synonymous variants in the EBV genome.** (a) The number of synonymous and non-synonymous variants in each open reading frame (ORF) is arranged by the total sum of variants in the ORF. The numbers are normalized by the number of individuals in each cohort and by the gene length per 1000 amino acids. (b) The number of synonymous and non-synonymous variants in latent and lytic genes. The numbers are further normalized by the total number of genes classified as latent/lytic. dN/dS, ratio of non-synonymous variant to synonymous variant.
(TIF)

**S4 Fig. Percentage of genotype call missingness along the EBV genome.** Each dot represents the percentage missingness of a variant. Repetitive regions in the EBV genome are shaded in yellow. Variants spanning the upstream of *EBER1* (immediately downstream of *BNRF1*) to the downstream of *EBER2* (immediately upstream of OriP), and *BALF2* are colored in red.
(TIF)

**S5 Fig. Distribution of samples carrying the risk alleles identified previously.** The two columns indicate the distribution of genotypes in the BALF2 risk loci and EBER-del respectively. The two rows indicate the distribution of genotypes of healthy controls and NPC cases respectively. P-values from the Fisher's exact test of allelic counts are shown on top of each bar.
(TIF)

**S6 Fig. Degree of linkage disequilibrium against p-values of meta-analysis.** Each dot represents a variant in the meta-analysis of GWAS.
(TIF)

**S7 Fig. Manhattan plot of conditional analyses.** (a) Conditional on EBER-del. (b) Conditional on variant 163364. (c) Conditional on variants 148489, 150028, 151822 and 152087. Repetitive regions in the EBV genome are shaded in yellow. Red line: Bonferroni-corrected genome-wide significance at $2.33 \times 10^{-05}$; blue line: permuted genome-wide significance at $2.35 \times 10^{-04}$.
(TIF)

**S8 Fig. Number of samples carrying the nine high risk SNPs in each cluster.** The number of cases/controls are labelled on each bar chart, colored according to their identity.
(TIF)

**S9 Fig. Performance metrics of 28 and 9 SNPs in predicting NPC.** Each boxplot represents the distribution of one performance metric, generated by a ten-fold cross-validation with five-repeats.
(TIF)

**S1 Table. Characteristics of 507 samples included in the current meta-analysis.**
(XLSX)

**S2 Table. Summary statistics for sequencing and variant calling for 507 samples.**
(XLSX)

**S3 Table. Variants that passed the permuted p-value cut-off in the meta-analysis of GWAS.**
(XLSX)

**S4 Table. Logistic regression of disease states on haplotypes constructed from 56 variants that passed p-value threshold in the meta-analysis, with age and sex as covariates.**
(XLSX)

**S5 Table. Distribution of cases and controls in the nine clusters.**
(XLSX)

**S6 Table. Genotypes of 38 SNPs that characterized the high-risk EBV lineage.**
(XLSX)

**S7 Table. Number of samples in each cluster carrying the 9 SNPs shared by the high-risk lineage only.**
(XLSX)

## Acknowledgments

We acknowledge Tissue Bank of NPC Area of Excellence (AoE/M 06/08 Centre of Nasopharyngeal Carcinoma Research) for providing additional NPC samples for the current study. Bioinformatic pipelines were executed on the High-Performance Computing Facility hosted by the Centre for PanorOmic Sciences of the University of Hong Kong. We thank TF Chan for his assistance on the experimental work. Publication was made possible in part by support from the HKU Libraries Open Access Author Fund sponsored by the HKU Libraries.

## Author Contributions

**Conceptualization:** Ka Wo Wong, Kwai Fung Hui, Alan K. S. Chiang.

**Data curation:** Ka Wo Wong, Kwai Fung Hui, Ki Pui Lam.

**Formal analysis:** Ka Wo Wong.

**Funding acquisition:** Alan K. S. Chiang.

**Investigation:** Ka Wo Wong, Kwai Fung Hui.

**Methodology:** Ka Wo Wong.

**Resources:** Dora Lai-wan Kwong, Maria Li Lung.

**Software:** Ka Wo Wong.

**Supervision:** Wanling Yang, Alan K. S. Chiang.

**Validation:** Ka Wo Wong, Kwai Fung Hui, Wanling Yang, Alan K. S. Chiang.

**Visualization:** Ka Wo Wong.

**Writing – original draft:** Ka Wo Wong, Wanling Yang, Alan K. S. Chiang.

**Writing – review & editing:** Ka Wo Wong, Kwai Fung Hui, Ki Pui Lam, Dora Lai-wan Kwong, Maria Li Lung, Wanling Yang, Alan K. S. Chiang.

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
