## [Decision Letter · Decision Letter 0]

26 Feb 2024

Dear Dr Chiang,

Thank you very much for submitting your manuscript "Meta-analysis of Epstein-Barr virus genomes in Southern Chinese identifies genetic variants and viral lineage associated with Nasopharyngeal Carcinoma" for consideration at PLOS Pathogens. As with all papers reviewed by the journal, your manuscript was reviewed by members of the editorial board and by several independent reviewers. In light of the reviews (below this email), we would like to invite the resubmission of a significantly-revised version that takes into account the reviewers' comments.

Both reviewers provide compelling concerns with the analysis. If the authors revise their study and meet these concerns, then their work and noted by one of the reviewers "the overall goal and much of the approach is exactly right and has been missing from other papers. "

We cannot make any decision about publication until we have seen the revised manuscript and your response to the reviewers' comments. Your revised manuscript is also likely to be sent to reviewers for further evaluation.

Sincerely,

Bill Sugden

Academic Editor

PLOS Pathogens

Patrick Hearing

Section Editor

PLOS Pathogens

Michael Malim

Editor-in-Chief

PLOS Pathogens

orcid.org/0000-0002-7699-2064

Both reviewers provide compelling concerns with the analysis. If the authors revise their study and meet these concerns, then their work and noted by one of the reviewers "the overall goal and much of the approach is exactly right and has been missing from other papers. "

Reviewer's Responses to Questions

**Part I - Summary**

Reviewer #1: Wong et al add a new set of case-control samples to two existing EBV genome datasets (one from their lab), and then extend existing SNP mapping to identify other risk variants. Importantly, this study is (to my knowledge) the first to extend these analyses to haplotype mapping, which is essential for moving towards the use of viral genetics to facilitate targeted screening. Doing so, they identify what appears to be a high risk haplotype.

While the goal and approach is generally right, I have two substantial criticisms that should be addressed before these results can be considered robust and informative. This essentially revolves around the way the analyses have been executed (or in some cases perhaps explained), in particular the lack of robust approaches to control for the unequal sizes of control and case groups from the two geographical areas being studied. Since I am not well versed in genetic methods, I cannot tell how difficult these would be to address.

Reviewer #2: Nasopharyngeal carcinoma (NPC) is prevalent in Southeast Asia. It is thought that environmental factors, human genetic variation, and genetic diversities of EBV contribute to EBV-associated NPC. Wong et al identified 38 single nucleotide polymorphisms (SNPs) in 61 NPC patients carrying high-risk EBV lineage which encodes a G6944A variant at the upstream of EBER2. However, if these SNPs indeed correlate with NPC incidences, they should be aligned to the sequence of M81 strain which was derived from an NPC patient rather than the B95.8 strain (Figure 5). All SNPs in the cluster 2, 3 and 7 should be listed, and create a Venn diagram of intersections for all SNPs. To conclude these G6944A and 38 SNPs can be implicated in the development of NPC, EBV strains from areas other than Southeast Asia should also be analyzed to ensure the specificity of these variations and SNPs.

**Part II – Major Issues: Key Experiments Required for Acceptance**

Reviewer #1: My major issue is the way the Hong-Kong and Guangzhou datasets were combined. The HK data set had a case: control ratio of 2:1 and the GZ data was a ratio of 1:2.5. Therefore any statistical analysis of SNP enrichment will also identify SNPs that statistically differ between HK and GZ. The authors present PCA that they claim shows the two populations are equivalent, but this is both not a suitable test for this (PCA includes only a subset of variation) and unconvincing (HK is significantly enriched in the bottom left quadrant of the PCA fig 2b, even if this simply reflects the more ethnically diverse history of HK). Therefore, the data would be much more robust if the authors used a two factor statistical test (maybe ANOVA) combining sampling site with case-control. It would also be helpful to present sample site-split case control data sets (and present proportions rather than case numbers) as supplementary to help validate data such as in figures 4b and 5b.

My second issue is around how the haplotype analysis was conducted (or explained). The study picked - apparently arbitrarily - 8 SNPs for their initial analysis. If this was all the mis-sense SNPs, then this makes the assumptions that any important SNPs will be coding (despite the fact that the top hit - EBER2 promoter) is clearly non-coding. The paper then produces a maximum risk haplotype of 38 SNPs using an agnostic method. In my view this has been done the wrong way around, and the agnostic method has not been leveraged sufficiently. The haplotype selection process is a bit opaque, and there is a lack of clarity of what percentage of cases have vs do not have this haplotype, and how to leverage the haplotype (and more importantly deviations from the haplotype) to inform predictive diagnosis. If this could be incorporated, it would maximise the value of the paper.

What would be really valuable to the field would be to know which are the biggest contributors to NPC within this high risk haplotype. I have seen cumulative contribution analyses when attempting to reduce gene signatures associated with a disease to the minimum size for maximum predictive value. My colleague tells me this is a machine learning strategy called “feature selection” (specifically forward selection - adding them one by one and checking the performance metrics [significance or odds ratio] each time), and/or using logistic regression or decision trees. I think this approach to selecting the most predictive SNPs would be an agnostic approach to both identifying the most predictive SNPs, and potentially identifying SNPs that would be most important for future biomarker studies. Such a refined set of SNPs could then be more rationally used in the analyses described in figure 4/tables 1&2. This kind of insight would make the paper both more robust (in terms of avoiding investigator assumptions) and more impactful in focusing future research.

Third, this sort of study is highly dependent on the multiple sequence alignment (MSA) used by the investigators. This MSA should be directly available, either from a data repository or as supplementary data file, to help others to validate the study.

Reviewer #2: (No Response)

**Part III – Minor Issues: Editorial and Data Presentation Modifications**

Reviewer #1: I also have a number of minor comments below that I compiled as read the paper, that should be considered or addressed, for clarity, accuracy, precision, or methodological robustness. This includes some references to the major issues mentioned above.

Abstract:

22 - clarify multiple SNPs (why variants?) forming a high risk haplotype? Or independent linkages?

26 Panel of SNPs: say something about interdependency/recombination/haplotype. Multiple mentions of panel (summary) seems redundant?

61 Line mentions preponderance for Type I in NPC without context of background type1:type2 prevalence in the high risk population. Comment on relative frequencies in South China.

64 - endemic regions: terminology confusing in the context of endemic BL. Regions of high NPC prevalence?

66. Ref 22 is much more complicated than the interpretation here: the paper made a chimeric M81/B95-8 EBV that likely resulted in mismatches between polymorphisms in N- and C-terminus, it may be these mismatches that reduced EBNA-1 function, rather than this polymorphism alone. It therefore may not reflect the real biology in circulating EBV strains.

70/71 - ref 28 shows SNPs common to China/indonesian EBVs (NPC-high regions) having altered miRNA expression profiles: it does not test NPC-association within those regions, so the claim should be toned down as a geographical link, rather than a disease link.

78-80 - when describing the EBER2 polymorphisms, it may be useful to the reader to include position relative to EBER1 (one of the two polymorphisms will be between EBER2 and EBER1) - eg “less than 20 nt upstream of EBER2”. Also, T6999G (from https://doi.org/10.1099/jgv.0.001728) position is within EBER2 on my NC7605 map. Please clarify (and clarify whether this is the functional EBER2 SNP characterised by the Delecluse (Li et al 2019) and Tibbetts (Wang et al 2022) labs).

94 - explain what “279 and 227” this was confusing: suggest attach the numbers to the descriptors. case vs control? [controls are assymmetrically matched?]

Statistics: How does the study mathematically handle the unequal sizes of cases and controls from each study? HK1 is 142+61/62+31; GZ is 61/156. How do you know your study is not showing differences between HK and GZ, because of this skew?

.

208 - Statistics conducted in R is insufficient information (unless the individual statistical tests are defined throughout the paper).

233-239 - Principal components are generated de novo for every data set, so the the implication that geography falls in PC2 or 3 (line 235-6) is not valid. Indeed, just looking at the PCA, you might (by the same logic) conclude that the NPC and non-NPC samples also are not different from each other. And looking at the data, there are no GZ samples (except one NPC) in the bottom part of the PCA. So I feel that the geography in the samples is not fully resolved. A more robust statistical test is required.

244 - ‘two cohorts’ - is this NPC/ctrl or HK/GZ?

250 - top association was replicated - was this replication strengthened, weakened or similar, given the additional samples (we would expect p to become even smaller with larger n).

254 - the non-replication of 163364 warrants additional scrutiny to explain the difference: is your re-calling the sequences changing the frequency of this SNP? Is the prevalence of this SNP different between HK and GZ samples? Or was this just skewed by the low control number in GZ data set the first time around?

267 - variants within R2 0.6-0.8 - this seems to be a different analysis from that in figure 3: Narratively the analysis that gives the degree of association should like to the p value measure in figure 3, or the figure indicating R2 values should be more prominent.

Overall - not sure I follow the logic of the narrative - surely if you are looking for driver risk alleles other than EBER, you want to investigate SNPs with high P value but low linkage, to maybe identify alternate haplotypes?

277-295 - a haplotype-based analysis is absolutely the right think to do, but the selection of the haplotype seems entirely subjective, and the basis for choosing these SNPs is unclear. Is it reasonable to select only coding mutations, as other (regulatory) changes could be more important but much harder to identify.

Table 1: REF and ALT may not be the best annotation - REF and NPC or REF/CHN (for China) may be better, since you are defining NPC-associated variants. In a table of all variants, then the “ALT” makes a bit more sense, although it is possible that a REF allele has higher association with NPC than ALT (where the OR is <1, presumably).

Table 2 is the crucial data set that shows a really strong enrichment - again I would prefer a supplemental version looking at the haplotype frequencies in the geographically separated data sets (GZ vs HK). However it is very hard to ‘read’ the haplotypes: could you have all the haplotypes labelled as bold (or coloured) for risk-allele and non-bold as no-risk allele. Then it is much easier to see both the number and structure of risk alleles.

I think it might also be useful to have a separate analysis (including rare alleles) that shows OR and/or p-value for different numbers of risk alleles from your list. Or if the haplotypes are more structured (showing recombination between two high and low risk haplotypes) then the total length of the risk haplotype.

In the 16 non-risk haplotype NPC cases, are there any risk-linked SNPs in common? Could these be assessed for being part of the haplotype, or an independent risk locus?

Thought: are there any CTL epitopes (esp in Chinese HLA) linked to the SNPs? Or even DRiPs associated with silent polymorphisms? [may not be possible to answer this]

Figure 4/317-324 - I am not clear how the clusters have been defined: They do not seem to make sense in the context of the phylogenetic tree (with cluster 3 being scattered among several parts of the tree) and the NPC risk-alleles. Perhaps it would have been more useful to cluster only using SNP positions that were defined as associated with NPC at some threshold, and place that in the wider phylogeny of the virus? Ether way, the authors need to explain the rationale behind their particular choice of clustering algorithm, as the results are hard to understand.

Figure 4b is not a great visualisation of the data - surely much better to use paired bars, with each bar indicating the % of cases or controls. This would offer a much easier to interpret visual representation of cluster enrichment in one or the other group.

Line 337 - the key observation (AUC value for ROC) is not easy for

---

## [Editor Report · Decision Letter 1]

9 May 2024

Dear Dr Chiang,

Thank you very much for submitting your manuscript "Meta-analysis of Epstein-Barr virus genomes in Southern Chinese identifies genetic variants and high risk viral lineage associated with nasopharyngeal carcinoma" for consideration at PLOS Pathogens. As with all papers reviewed by the journal, your manuscript was reviewed by members of the editorial board and by several independent reviewers. The reviewers appreciated the attention to an important topic. Based on the reviews, we are likely to accept this manuscript for publication, providing that you modify the manuscript according to the review recommendations.

Your responses to the first reviewer are clear. Those to the second reviewer are a bit confusing and need clarification. In particular: "The authors seem to recognize that their analysis requires them first to identify which strain of EBV they consider (type I or type II) and then identify sequence variants in their isolates from each strain that differ form their type I or type II reference genome. However, their description of their analysis is unclear. For example, they write: " To determine the EBV type of our samples, an EBV Type II BAM file was generated by an alignment of the Type II EBV reference genome AG876 (GenBank accession number: NC_009334.1) to the Type I EBV reference genome. The BAM file for each sample was piled up together with the Type II EBV reference BAM file, and variants were called by BCFtools (version 1.13). Nucleotides that were identical in both Type I and Type II EBV genomes were ignored." This description does not make sense and needs to be corrected so that a reader can understand the steps in the authors' analysis to ensure that it does make sense." Can you please revise this section of your manuscript to address this need?

Sincerely,

Bill Sugden

Academic Editor

PLOS Pathogens

Patrick Hearing

Section Editor

PLOS Pathogens

Michael Malim

Editor-in-Chief

PLOS Pathogens

orcid.org/0000-0002-7699-2064

Your responses to the first reviewer are clear. Those to the second reviewer are a bit confusing and need clarification. In particular: "The authors seem to recognize that their analysis requires them first to identify which strain of EBV they consider (type I or type II) and then identify sequence variants in their isolates from each strain that differ form their type I or type II reference genome. However, their description of their analysis is unclear. For example, they write: " To determine the EBV type of our samples, an EBV Type II BAM file was generated by an alignment of the Type II EBV reference genome AG876 (GenBank accession number: NC_009334.1) to the Type I EBV reference genome. The BAM file for each sample was piled up together with the Type II EBV reference BAM file, and variants were called by BCFtools (version 1.13). Nucleotides that were identical in both Type I and Type II EBV genomes were ignored." This description does not make sense and needs to be corrected so that a reader can understand the steps in the authors' analysis to ensure that it does make sense." Can you please revise this section of your manuscript to address this need?

Reviewer Comments (if any, and for reference):

Figure Files:

Data Requirements:

Reproducibility:

References:

---

## [Editor Report · Decision Letter 2]

15 May 2024

Dear Dr Chiang,

We are pleased to inform you that your manuscript 'Meta-analysis of Epstein-Barr virus genomes in Southern Chinese identifies genetic variants and high risk viral lineage associated with nasopharyngeal carcinoma' has been provisionally accepted for publication in PLOS Pathogens.

Best regards,

Bill Sugden

Academic Editor

PLOS Pathogens

Patrick Hearing

Section Editor

PLOS Pathogens

Michael Malim

Editor-in-Chief

PLOS Pathogens

orcid.org/0000-0002-7699-2064
---

## [Editor Report · Acceptance letter]

22 May 2024

Dear Dr Chiang,

We are delighted to inform you that your manuscript, "Meta-analysis of Epstein-Barr virus genomes in Southern Chinese identifies genetic variants and high risk viral lineage associated with nasopharyngeal carcinoma," has been formally accepted for publication in PLOS Pathogens.

Best regards,

Michael Malim

Editor-in-Chief

PLOS Pathogens

orcid.org/0000-0002-7699-2064